# ShcD Binds DOCK4, Promotes Ameboid Motility and Metastasis Dissemination, Predicting Poor Prognosis in Melanoma

**DOI:** 10.3390/cancers12113366

**Published:** 2020-11-13

**Authors:** Ewa Aladowicz, Letizia Granieri, Federica Marocchi, Simona Punzi, Giuseppina Giardina, Pier Francesco Ferrucci, Giovanni Mazzarol, Maria Capra, Giuseppe Viale, Stefano Confalonieri, Sara Gandini, Fiorenza Lotti, Luisa Lanfrancone

**Affiliations:** 1Sarcoma Molecular Pathology Team, Divisions of Molecular Pathology and Cancer Therapeutics, The Institute of Cancer Research (ICR), 15 Cotswold Road, Sutton SM2 5NG, UK; ewa.aladowicz@icr.ac.uk; 2Department of Experimental Oncology, European Institute of Oncology IRCCS (IEO), Via Adamello, 16, 20139 Milan, Italy; letizia.granieri@ieo.it (L.G.); federica.marocchi@ieo.it (F.M.); simona.punzi@ieo.it (S.P.); giuseppina.giardina@ieo.it (G.G.); pier.ferrucci@ieo.it (P.F.F.); stefano.confalonieri@ieo.it (S.C.); sara.gandini@ieo.it (S.G.); 3Division of Pathology, European Institute of Oncology IRCCS (IEO), Via Ripamonti 435, 20141 Milan, Italy; giovanni.mazzarol@ieo.it (G.M.); maria.capra@ieo.it (M.C.); giuseppe.viale@ieo.it (G.V.); 4Department of Oncology and Hemato-oncology, University of Milan, 20141 Milan, Italy; 5Candiolo Cancer Institute—FPO IRCCS, 10060 Candiolo Torino, Italy

**Keywords:** melanoma metastasis, ShcD adaptor protein, amoeboid motility, Rac1, DOCK4, melanoma PDX

## Abstract

**Simple Summary:**

Metastasis formation and dissemination is a complex process that relies on several steps. Even though highly inefficient, metastasis spreading is the primary cause of cancer morbidity and *mortality* in patients. The aim of our study was to investigate the molecular pathways leading to metastases making use of human-in-mouse melanoma models of patient-derived xenografts. We demonstrate that the modulation of the expression of an adaptor protein of the Shc family, ShcD, can change the phenotype and the invasive properties of melanoma cells when highly expressed. We also show that ShcD binds DOCK4 and confines it into the cytoplasm, blocking the Rac1 signaling pathways, thus leading to metastasis development. Moreover, our results indicate that melanoma cells are more sensitive to therapeutic treatments when the ShcD molecular pathway is inactivated, suggesting that new therapeutic strategies can be designed in melanomas.

**Abstract:**

Metastases are the primary cause of cancer-related deaths. The underlying molecular and biological mechanisms remain, however, elusive, thus preventing the design of specific therapies. In melanomas, the metastatic process is influenced by the acquisition of metastasis-associated mutational and epigenetic traits and the activation of metastatic-specific signaling pathways in the primary melanoma. In the current study, we investigated the role of an adaptor protein of the Shc family (ShcD) in the acquisition of metastatic properties by melanoma cells, exploiting our cohort of patient-derived xenografts (PDXs). We provide evidence that the depletion of ShcD expression increases a spread cell shape and the capability of melanoma cells to attach to the extracellular matrix while its overexpression switches their morphology from elongated to rounded on 3D matrices, enhances cells’ invasive phenotype, as observed on collagen gel, and favors metastasis formation in vivo. ShcD overexpression sustains amoeboid movement in melanoma cells, by suppressing the Rac1 signaling pathway through the confinement of DOCK4 in the cytoplasm. Inactivation of the ShcD signaling pathway makes melanoma cells more sensitive to therapeutic treatments. Consistently, ShcD expression predicts poor outcome in a cohort of 183 primary melanoma patients.

## 1. Introduction

Over the last decade, the worldwide incidence of cutaneous melanoma raised more rapidly than that of any other cancer type [1]. Though early-stage melanoma is curable by surgical excision in most cases, when tumor thickness increases, melanomas tend to spread and colonize other organs, leading to dismal prognosis.

Melanoma metastatic dissemination is a complex process primarily involving the phenotypic plasticity of melanoma cells, a process that includes adaptive modifications of gene expression, changes of cell morphology, detachment from the extracellular matrix (ECM), loss of cell–cell contacts and spreading [2,3]. Though the metastatic process is per se highly inefficient, in melanomas it is favored by the high plasticity and strong motility of melanoma cells [3,4]

Depending on the matrix composition of the tumor microenvironment, migrating melanoma cells can adopt two mutually exclusive phenotypes, corresponding to different modalities of motility, either the so-called rounded (amoeboid) or the elongated (mesenchymal) phenotype, and can switch between them [4,5]. Both phenotypes are driven by regulators of actin contractility belonging to the Rho family of small GTPases [6,7]. The rapid, amoeboid motility is regulated by RhoA, and is based on high actomyosin contractility enabling cells to squeeze through the matrix. It is characterized by weak or no adhesion to ECM, with little or no proteolysis of the matrix, and is driven by membrane blebs, actin-rich pseudopodia and/or highly contractile uropods [8,9,10,11]. On the contrary, elongated movement, characterized by actin assembly, cell polarization and the digestion of an extracellular matrix by proteases, is driven by Rac1 [6]. These two pathways show inhibitory effects on each other. Notably, amoeboid movement is prominent at the invasive front of melanoma in animal models [6,12,13], as well as in human melanoma lesions [7,13],suggesting that it may favor tumor invasion.

The switch between the amoeboid and elongated movements is based on the activation of the small GTPases cycle [14]. In melanoma, it was previously shown that DOCK3, a Rac1-specific guanine nucleotide exchange factor (GEF), is required for mesenchymal movement, whereas the Rac1 suppressor ARHGAP22 has an inhibitory effect. [12,14]. Another member of the DOCK GEF family, DOCK10, drives the amoeboid motility of melanoma cells through the activation of Cdc42, another Rho-family GEF [15]. DOCK4 expression has been previously linked to cell migration in vitro and to the risk of developing bone metastasis in breast cancer patients [16,17,18] as well as correct axon guidance in vivo during embryo development [17], thus suggesting a prominent role of DOCK4 in regulating cell motility.

ShcD/Shc4/RaLP is a cytosolic protein belonging to the SHC adaptors family. During mouse development, this plays a critical role in the transition from embryonic to epiblast stem cells; in the adult it is expressed in the developing nervous system, heart, muscle and skin [19]. In humans, ShcD expression is restricted to invasive and metastatic melanomas, and its silencing reduces cancer cell migration [20]. ShcD phosphorylation on specific tyrosine residues by activated receptor tyrosine kinases leads to the transient stimulation of MAPK signaling. MAPK activation by phosphorylated ShcD, however, is not sufficient to support the migration of metastatic melanomas, suggesting that ShcD activates other critical, MAPK-independent migratory pathways [20].

Here, we show that the modulation of ShcD protein levels affects the cell morphology and invasiveness in melanoma patient-derived xenografts (PDXs) [21]. ShcD expressing cells detach from ECM, acquire rounded morphology and invasive traits, as observed in in vitro assays on collagen substrates invading collagen. Moreover, in three-dimensional (3D) assays, cells switch from a mesenchymal to amoeboid phenotype. In vivo amoeboid ShcD overexpressing cells disseminate to lymph nodes and seed distant metastases. Consistently, analyses of a large cohort of melanoma patients demonstrated that ShcD is a prognostic factor in melanoma. Notably, ShcD silencing sensitizes BRAF-mutant melanoma cells to targeted therapy [22,23], suggesting that the regulation of ShcD expression influences both the cell motility and drug sensitivity of melanoma cells.

## 2. Results

### 2.1. ShcD Impairs the Adhesive Properties of Melanoma Cells

Using metastatic melanoma cell lines (WM266-4 and IGR-37), we previously showed that decreased ShcD expression reduces migration without interfering with cell proliferation in vitro [20]. We confirmed these data in PDX cells (MM27) by the lentiviral transduction of ShcD-targeting shRNAs (shShcD#1 and shShcD#2 pooled together; ShShcD) or control (ShLuc) vectors (Appendix A). Since tumor cell migration involves modifications of cell- and stroma-interactions, we analyzed whether ShcD regulated the adhesive properties of MM27 PDX cells. ShShcD and ShLuc cells were plated on different extracellular matrices, including fibronectin, collagen and matrigel, and counted by Crystal Violet at different time points after the extensive wash-out of floating cells. As shown in Figure 1A, ShcD depletion significantly increased numbers of adherent cells to all tested matrices. To visualize the morphology of ShShcD MM27 adherent cells, cell spreading was analyzed upon adhesion to fibronectin. A higher percentage of spread cells was observed in ShShcD cells, as compared to control shLuc cells (Figure 1B), indicating that ShcD silencing increases cell attachment and spreading capacities.

Cell spreading depends on the formation of focal adhesions (FA), multi-protein complexes that serve to connect the cellular cytoskeleton with components of the extracellular matrix. We analyzed FA formation by staining MM27 cells with antibodies against known components of the complex, e.g., vinculin, paxillin and focal adhesion kinase (FAK) (Appendix A) and their phosphorylated counterparts (Figure 1C) [24]. After adhesion to fibronectin, we observed a significant increase in the number and intensity of phospho-FA staining in ShcD knockdown cells (Figure 1C). Together, these results demonstrate that ShcD impairs the ability of melanoma cells to adhere to extracellular matrix components, through the modulation of FA formation, thus favoring cell migration.

### 2.2. ShcD Regulates Melanoma Cell Morphology and Sustains Amoeboid Movement of Melanoma Cells in 3D Matrix

The capacity of melanoma cells to switch to different morphologies can be visualized in vitro by culturing cells in 3D matrix conditions. We first analyzed the morphology of MM27 PDX cells overexpressing ShcD plated on thick collagen layers (Figure 2A). While control cells (PincoPuro (PP)-vector) showed mixed morphologies when plated on thick collagen layers (65% rounded and 35% elongated) (Figure 2B), rounded cells raised to 87% in ShcD overexpressing cells (PP-ShcD), suggesting that ShcD drives morphological changes in melanoma cells. Similar results were obtained in WM115 and WM266.4 cells (Appendix A), two independent cell lines isolated, respectively, from the primary and metastatic tumors of the same patient. Both cell lines were transduced with a control vector (ShLuc), shShcD#1 and shShcD#2 vectors. The WM115 cell line consists mainly of rounded cells (79%), while WM266.4 is composed of a mixed population of rounded and elongated cells, as in the MM27 PDX. In WM115, ShcD silencing decreased the population of rounded cells to 27% for shShcD#1 (*p* < 0.0001) and 48% for shShcD#2 (*p* < 0.0001) (Appendix A). Similarly, in WM266.4, ShcD silencing reduced rounded cells to 38% and 42% (*p* < 0.001) with shShcD#1 and shShcD#2, respectively. Morphology of elongated cells within the two ShcD-interfered melanoma cell populations is shown in Appendix A.

We then investigated whether ShcD influences melanoma cell invasion using melanoma spheroids embedded in collagen gel. As shown in Figure 2C, ShcD overexpression increased significantly the invasive properties of MM27 cells, as compared to the control. Both control (PP) and ShcD overexpressing cells (PP-ShcD) were capable of invading the matrix over time, but the PP-ShcD cells resulted as significantly more invasive than the control cells at all time points (Figure 2C). Notably, time-lapse analyses showed that invasive PP-ShcD cells move faster than the controls (Figure 2D, Appendix A).

Melanoma cells invade the surrounding tissue either collectively or as single-cells [25]. When tumor cells detach from ECM, they start invading the surrounding tissue as single cells and switch to rounded/amoeboid movement [25]. The amoeboid phenotype allows cells to rapidly move out of the tumor: indeed, amoeboid cells are found at the invasive fronts of primary tumors close to tumor-associated macrophages and vessels [26]. We therefore analyzed the invasive front of melanoma spheroids in 3D assays. The invasion of control (PP) and ShcD overexpressing (PP-ShcD) collagen-embedded PDX cells were visualized by time-lapse for 24 h. We observed significantly increased numbers of amoeboid cells among the ShcD overexpressing cells (Figure 2E, Appendix A).

These data indicate that ShcD expression favors the rounded morphology of melanoma cells on thick collagen layers, adhesion, migration and invasion in vitro, suggesting that ShcD is a critical determinant of the plasticity and migratory movement of melanoma cells.

### 2.3. ShcD Is Crucial for Metastasis Dissemination

To investigate the potential of ShcD in promoting metastasis formation in vivo, MM13 and MM27 PDXs were chosen as representatives of melanoma subtypes with invasive or proliferative phenotypes, respectively [21]. MM13 PDX carries the NRAS mutation and displays a more invasive phenotype, with high levels of AXL, PDGF-Rβ, EGFR and EPHA2 proteins and very low levels of MITF, Brn2 and SOX10. MM27, instead, harbors the mutated BRAF gene and expresses MITF, BRN2 and SOX10, thus suggesting a proliferative phenotype [27] (Figure 3A). The retroviral transduction of MM13 and MM27 PDX cells with the PP-ShcD vector allowed for high levels of ShcD expression in both PDXs, as compared to the PP-vector transduced cells (Figure 3B). Analyses of the migratory potential of SchD overexpressing MM13 and MM27 cells showed increased migratory capability (Figure 3C). To investigate the metastatic potential of ShcD overexpressing cells, transduced cells were injected into the back dermis of five NSG mice and the tumors were surgically removed to prolong animal survival and allow for metastasis detection. Under these experimental conditions, we showed that MM13 and MM27 can metastasize to different organs (including lymph nodes, lung, liver and spleen) (Figure 3C–H). Mean latency to a volume of the primary tumor of 0.35–0.5 cm^3^ was similar in the two PDXs, ranging from 21 to 25 days (MM27 and MM13, respectively), with no statistically significant difference with the control PP-vector tumors (Figure 3D). At early time points, after the surgical resection of the primary tumor (approx. 1 month), both PP and PP-ShcD mice developed metastases at lymph nodes, with comparable frequencies (4/5 vs. 5/5 in the MM13 group; 2/2 vs. 2/2 in the MM27 group). However, the size of the metastatic lymph nodes was significantly enlarged in the MM13 PP-ShcD mice (~4–5 folds; Figure 3E, upper panel; *p* < 0.0001). A similar trend was observed in MM27 (~3–4 folds; Figure 3E, lower panel), though the small number of mice developing metastases did not enable a statistical representation.

We then investigated whether ShcD overexpression led to metastasis formation at distant organs, by observing the animals up to 2 months from the resection of the primary tumor and using MM13 as model. As shown in Figure 3F, lymph nodes, lung, liver and spleen were colonized by PDX cells. At 2 months, the size of the metastatic lymph nodes was significantly enlarged in the MM13 PP-ShcD mice, similar to what we had previously observed at earlier time points (Figure 3G). Most notably, the numbers of metastatic events, regardless of the specific sites, were markedly higher in PP-ShcD mice, as compared to the controls (seven independent metastatic events in the seven control mice; 16 in the seven PP-ShcD mice; *p* = 0.0298) (Figure 3H). Thus, ShcD may activate molecular pathways that are crucial for the metastasization of melanoma cells.

### 2.4. ShcD Expression Correlates with Melanoma Progression and Patient Outcome in Melanoma Patients

We sought to interrogate how the ShcD RNA transcript is regulated in a cohort of primary and metastatic melanomas. A total of 256 archival paraffin-embedded melanoma patients were included in the study, 183 patients carrying primary melanomas and 94 melanoma metastatic tissues (Table 1). We could retrieve matched primary and metastatic samples from 21 patients. The ShcD positive rate was significantly higher in the metastatic tissue than in primary melanoma (54% and 34% for primary and metastatic tissues, respectively; *p* = 0.0003), as previously shown in a small cohort of patients [20].

Demographics, as well as the clinical and histo-pathologic features of the melanoma patients and follow-up events are listed in Table 1. The ShcD RNA transcript showed statistically significant associations with all the main prognostic factors. ShcD-positive melanomas were significantly thicker (Breslow’s thickness >4 mm was 35% among ShcD-positive vs. 8% among ShcD-negative; *p* < 0.0001 overall), with more frequent ulceration (51% among ShcD-positive vs. 12% among ShcD-negative; *p* < 0.0001), with a greater number of mitotic events (94% among ShcD-positive vs. 46% among ShcD-negative patients; *p* < 0.0001) and they were more often nodular melanomas (46% in ShcD-positive vs. 20% in ShcD-negative; *p* = 0.001). ShcD expression was also associated with the stage (*p* = 0.0004) and lymph nodes status (*p* = 0.02): the majority of advanced stage patients were ShcD-positive (30% in stage III and IV vs. 15%, *p* = 0.0004)**,** and the frequency of positive lymph nodes was significantly greater in the positive (28%) rather than negative (14%) ShcD expressing patients (*p* = 0.02). Multivariate logistic models, adjusted for confounders, confirmed these associations. These findings strengthen the association of ShcD expression with the increased aggressiveness of the tumor.

Overall, with a median follow-up of 100 months, 35% of the patients within our cohort of primary melanoma patients died, and we found a significant worse overall survival (log-rank *p* = 0.05) among the ShcD-positive as compared to the ShcD-negative patients: at two years, we had 52% vs. 72% probability of survival, respectively (Figure 4A). Disease-free survival at two years was also significantly worse (66%) in ShcD positive than in ShcD-negative patients (78%; log-rank test *p* = 0.04, Figure 4B; stage IV patients were not included in this analysis). The Cox regression model, adjusted for age and gender, indicated a trend toward a significant association of ShcD expression with disease-free survival (*p* = 0.08).

ShcD expression was then evaluated in metastatic melanoma tissue in 94 patients (Table 1). The analysis of the prognostic factors in the antecedent melanoma showed no significant association with ShcD expression. Of note, ShcD-positive patients were younger than the ShcD-negative (median age at diagnosis 45 vs. 50; *p* = 0.02), more frequently carrying ulcerated melanomas (*p* = 0.04). In these patients, ShcD expression is associated with significantly more distant metastases (*p* = 0.01), suggesting that ShcD might predict for metastasis dissemination.

Comparison between the evaluation of ShcD in primary and metastatic tissues confirms the hypothesis that ShcD is a prognostic marker, even if probably not independent from other prognostic markers, and that it is associated with an unfavorable outcome in melanoma patients.

### 2.5. Rac1 Molecular Pathway Is Altered in ShcD Expressing Melanoma Cells

The metastatic process is mainly driven by aberrant cell migration, which relies on specific actin cytoskeleton dynamics and cell phenotypes activated by microenvironmental signals. In melanoma, the elongated/mesenchymal phenotype of cells is driven by activated Rac1 [6]. As shown in Appendix A, ShcD silencing switches the morphology of WM115 cells from rounded to elongated. To investigate whether the elongated phenotype of ShcD-depleted cells depends on Rac1, we either treated WM115 cells with the NSC23766 Rac1 specific-inhibitor [28] or we silenced Rac1 by means of Rac1 SmartPool siRNA (Appendix A). As expected, NSC23766-treated control WM115 cells or Rac1-silenced cells showed significantly increased numbers of rounded cells (*p* < 0.01) (Figure 5A,B). Notably, NSC23766 treatment rescued the rounded morphology of ShcD-silenced cells (from 28% to 61% for shShcD#1 and from 46% to 75% for shShcD#2 (Figure 5A)), demonstrating that cell elongation after ShcD knockdown is Rac1-dependent and suggesting that ShcD acts as a suppressor of the Rac1 pathway.

We therefore investigated whether the downregulation of ShcD leads to Rac1 activation. Pull-down assays of Rac1-GTP in WM115 cells upon adhesion to fibronectin showed higher levels of Rac1-GTP in ShcD knockdown cells, as compared to controls (Figure 5C). Consistently, the phosphorylation levels of N-WASP and Cofilin, two Rac1 downstream effectors [29], were markedly increased in ShcD downregulated cells (Figure 5D), demonstrating that ShcD silencing leads to an increased activation of the Rac1 signaling pathway.

### 2.6. ShcD Negatively Regulates Rac1 Activation by Preventing Its Binding to DOCK4

Rac1 is activated by different upstream regulators, including DOCK4, a guanine nucleotide exchange factor (GEF) GTPase that promotes Rac-dependent cell migration [18]. To investigate its relationship with ShcD, we assessed the presence of ShcD and DOCK4 in the same protein complex, using ShcD overexpressing MM27 PDX cells. As shown in Figure 6A, ShcD protein was detected in the DOCK4 immuno-precipitate. These data were also confirmed in ShcD overexpressing WM115 cells (Appendix A).

Based on the demonstrated ShcD–DOCK4 interaction and the role of DOCK4 in mediating Rac1 activation, we hypothesized that ShcD may negatively regulate Rac1 activation by preventing its binding to DOCK4. To test this hypothesis, we analyzed DOCK4 localization in the control and ShcD-silenced MM27 and WM115 cells after adhesion to fibronectin by immunofluorescence. In shLuc control cells, DOCK4 was found in the cytoplasm, while in ShShcD cells, DOCK4 was also partly localized at the plasma membrane (Appendix A). To precisely quantify the percentage of DOCK4 confined at the plasma membrane, the control and ShcD-silenced MM27 were co-stained with DOCK4 and CD146, a known cell adhesion molecule present on the plasma membrane of melanoma cells (Figure 6B). All together, these results suggest that when ShcD is overexpressed, it sequesters DOCK4 in the cytoplasm, thus blocking its recruitment to the plasma-membrane where it would activate Rac1. To strengthen the molecular value of our model, we silenced DOCK4 in MM27 cells using SmartPool siRNA. Once the silencing was confirmed by Western blot (Appendix A), we plated the DOCK4-silenced, control, and ShcD-silenced cells on collagen and we evaluated the effect of the silencing on cell morphology (Figure 6C). DOCK4-silenced cells recapitulated the elongated morphology of ShcD-silenced cells confirming the ShcD role in melanoma cell invasiveness through DOCK4 regulation.

### 2.7. ShcD Contributes to Resistance of Melanoma Cells to Treatment

Highly invasive cells are frequently resistant to standard care therapy. To investigate whether ShcD expression modulates the drug sensitivity of melanoma cells, MM27 PDX cells, carrying the BRAF^V600E^ mutation, were transduced with ShLuc and ShShcD vectors and treated with Dabrafenib (BRAF inhibitor), Trametinib (MEK inhibitor) or Everolimus (mTOR inhibitor). Analyses of cell viability at three different doses of the three drugs showed no difference between the control and ShcD-silenced cells (Figure 7A). We then analyzed the effects on the migration of sub-apoptotic doses of each of the three drugs. As expected, the reduction in ShcD protein levels lowered the migration rate in untreated cells (Appendix A). The rate of migration was significantly affected by all three drugs in the control cells and to a similar extent, also in ShcD-silenced cells (Figure 7B). Finally, we tested the effects on the migration of combinations of Dabrafenib, Trametinib and Everolimus. Treatment with the combination of Dabrafenib and Trametinib (Dab/Trame; a standard treatment in BRAF-mutant melanomas) had no effect on the migration of ShShcD cells compared to the control cells, showing that the drug response to the inhibition of the BRAF/MEK pathway is not influenced by the depletion of ShcD. Notably, treatment with the combination of Dabrafenib and Everolimus (Dab/Eve; a new combination therapy used to bypass the induction of resistance to BRAF inhibitors, [30]) significantly reduced the migration of ShShcD MM27 cells, as compared to the ShLuc control cells (Figure 7B). This indicates that the ShcD depletion increases the drug sensitivity of melanoma cells to this combination. All together, these data suggest that ShcD affects intracellular pathways that can, in turn, sensitize melanoma cells to new combination therapies.

## 3. Discussion

Melanoma cells are characterized by a high propensity to migrate and metastasize, which is probably linked to the embryologic derivation of its cell of origin. Cutaneous melanoma is derived from melanocytes, which originate from the neural crest. During development, cells exiting from the neural crest undergo morphological changes accompanied by altered adhesion, increased migratory capacity and spreading throughout the embryo before they reach target tissues and differentiate. The process of melanoma migration and spreading is associated with the intrinsic properties of neural crest-derived melanocytes and is tightly regulated by the surrounding environment. Notably, metastatic melanoma cells can be reprogrammed to neural crest-like cells by the embryonic microenvironment [31]. Moreover, during the invasion of the epidermal/dermal layer of the skin, melanoma cells go through an EMT-like program and at least partially exploit the same migratory transcriptional program used by neural crest cells during the development to colonize the skin [32]. Most notably, ShcD is expressed in neural crest cells [33] and melanoblasts [34], thus suggesting that it represents a lineage-specific factor critical for the migration of melanocyte precursors, whose expression and function can be reacquired by invasive melanoma cells.

We previously showed that ShcD positively regulates melanoma cell migration in the 2D cultures of melanoma cell lines [19]. Here, firstly, we showed that PDX models, which more faithfully phenocopy patients’ biological features, recapitulate the ShcD migratory function formerly documented in cell lines (Appendix A). Furthermore, we extended our analyses and demonstrated that ShcD silencing increased the capability of melanoma cells to adhere to different extracellular matrices, suggesting that ShcD expression favors the detachment of melanoma cells from the surrounding microenvironment, thus favoring dissemination (Figure 1).

During the last decade, the invasive properties of melanoma cells have been extensively studied using 3D model systems, which better mimic the tumor environment, as compared to the traditional 2D cell cultures. Most notably, our 3D approach showed that invasive melanoma cells are featured by high plasticity, with cells capable of modulating either RhoA or Rac1 small GTPases and thus adopting two alternative modes of invasion: amoeboid (rounded) or mesenchymal (elongated), respectively [6,8]. Interestingly, the MITF expression in the WM266.4 melanoma cell line increased the numbers of elongated cells and decreased their invasiveness, while MITF silencing in 501Mel cells induced rounded morphology, consistently with the capacity of MITF to negatively regulate the invasive phenotype of melanoma cells [35]. Consistently, MITF has been reported to downregulate ShcD expression by directly binding to at least five sites of the genomic locus [36], suggesting that the MITF/ShcD axis is a critical regulator of invasiveness phenotypes of melanoma cells. Notably, ShcD overexpression increased the invasion properties of melanoma cells in 3D assays (Figure 2). Since adhesion, in vitro migration and invasion are all pivotal steps in metastasis formation in vivo, we analyzed the metastatic properties of ShcD overexpressing cells in two different PDX models and showed that ShcD endowed melanoma cells with high metastatic potential (Figure 3). Notably, the metastatic tumor burden was increased by ShcD, as lymph nodes display a larger volume and a higher number of distant metastases is detected throughout the body of the animal. The metastatic tumor burden is significantly associated with the clinical response and outcomes in melanoma patients [37]. Not surprisingly, we showed that high ShcD expression is associated with a poor outcome in a cohort of melanoma patients (Figure 4).

The mechanisms regulating the different modalities of cell motility involve several signaling pathways. RhoA signaling activates ARHGAP22 Rho-GAP, which keeps Rac1 in an inactive state leading to high actomyosin contractility [6]. On the other hand, RhoA signaling is inhibited by WAVE2, which is activated by Rac1 [6]. Consistently, cells that adopt the mesenchymal mode of motility are characterized by the high activity of Rac1 [6]. The treatment of ShcD-knockdown cells with the selective Rac1 inhibitor NSC23766, as well as the silencing of Rac1, rescued the rounded phenotype of melanoma cells (Figure 5A,B), indicating that ShcD actively inhibits Rac1, inducing the rounded morphology of melanoma cells. In fact, ShcD-silenced cells showed increased Rac1 activation and elongated morphology. Based on these results, we hypothesized that ShcD inhibited the Rac1 signaling pathway, enabling active RhoA to guide amoeboid motility. Indeed, Rac1 inhibition in ShcD-knockdown cells reverted them to an amoeboid phenotype, suggesting that RhoA was active and did not need ShcD to execute its functions. We investigated the mechanism through which ShcD blocks Rac1 activation and we showed that ShcD formed a stable complex with DOCK4 (Figure 6A), a protein belonging to the DOCK180-related family of guanine nucleotide exchange factors (GEFs), whose function was already linked to cell invasion [16]. All Shc family members are equipped with different domains (CH2, PTB, CH1 and SH2) that enable them to recruit different signaling molecules to properly convey and organize the extracellular signals and that, at the same time, allow them to translate signals into specific biological responses [38]. ShcD differs from the other members of the family being selectively expressed in the neuronal-derived tissues only [38]. The specificity of expression suggests that ShcD drives crucial functions in these cells. Here, we demonstrate for the first time a direct binding of ShcD with the guanine nucleotide exchange factor DOCK4 and we show that after ShcD silencing, DOCK4 is recruited to the plasma membrane during cell adhesion to fibronectin (Figure 6B). In the same experimental conditions, Rac1 was highly activated. Conversely, in the presence of ShcD, the translocation of DOCK4 to the membrane did not occur and Rac1 was less active than in the ShcD knockdown cells. The recruitment of DOCKs to the membrane was previously shown to be an important step in the activation of Rac1 after the adhesion of epithelial cells to fibronectin or stimulation with growth factors [39,40]. Our results suggest that in our model system, ShcD prevents Rac1 activation by blocking DOCK4 recruitment to the plasma membrane (by a not yet identified mechanism). This would be an additional mechanism to keep the RhoA signaling pathway activated and promote amoeboid motility.

Having shown that ShcD overexpressing cells migrate and invade the surrounding tissue and metastasize to lymph nodes and distant organs, we sought to investigate whether ShcD depletion correlates with an increased sensitivity to targeted therapy. To this end, we used a BRAF-mutated melanoma (MM27). In patients, these melanomas are mostly treated with combination therapy based on the use of BRAF/MEK inhibitors, but the duration of response is variable and patients frequently experience recurrence: new treatment strategies, especially different combinations, are therefore needed. It was previously shown that Dabrafenib/Everolimus combination was effective in melanoma cell lines to bypass the induction of resistance to BRAF inhibitors [30]. Here, we report that this combination can become even more efficacious by preventing ShcD activation. Indeed, ShcD depletion does not influence the MAPK intracellular pathway [20], in line with our finding that shShcD cells are as sensitive to the standard of care therapy (Dabrafenib + Tramentib) as control cells, but it does activate the Rac1 signaling pathway, which in turn activates the mTOR pathway [41]. The upregulation of mTOR signaling makes melanoma cells more sensitive to Everolimus, thus suggesting that ShcD depletion can potentiate the effect of combined targeted therapy in melanoma cells.

Our findings demonstrate a key role of the adaptor protein ShcD in the process of metastasis formation and drug resistance. We propose to take into consideration ShcD as a putative therapeutic target to improve the prognosis of melanoma patients.

## 4. Materials and Methods

### 4.1. Plasmids, Lentiviral Transduction and Cell Culture

#### 4.1.1. Plasmids

Human ShcD cDNA was cloned in the Pinco plasmid [42]. The original backbone was modified to a new version, PincoPuro (PP) plasmid, containing the puromycin cassette under the 5′LTR and GFP (PP, empty vector) or ShcD cDNA (PP-ShcD, ShcD overexpressing vector) under the CMV promoter. LL3.7 Puro2GFP (LLPG) plasmid and its modified version with shRNA targeting Luciferase (shLuc) were a generous gift of Dr. Bruno Amati. Two different short hairpin RNAs (shRNAs) targeting ShcD (shShcD#1 and shShcD#2) were inserted in the XbaI and NotI restriction site of the LLPG plasmid. The following pairs of shRNA sequences were annealed and cloned into the vector backbone:

shShcD#1 forward 5′TCAATGAGATCACTGGATTTTTCAAGAGAAAATCCAGTGATCTCATTGTTTTTTC3′; shShcD# reverse 5′TCGAGAAAAAACAATGAGATCACTGGATTTTCTCTTGAAAAATCCAGTGATCTCATTGA3′;

shShcD#2 forward 5′TTGAGGAGGTGCATATTGATTTCAAGAGAATCAATATGCACCTCCTCATTTTTTC3′; shShcD#2 reverse 5′TCGAGAAAAAATGAGGAGGTGCATATTGATTCTCTTGAAATCAATATGCACCTCCTCAA3′

#### 4.1.2. Retroviral and Lentiviral Transfections and Infections

Phoenix-AMPHO helper cell line was transfected with PincoPuro constructs for retroviral production as previously described [42]. Three cycles of infection (3 h each) with supernatant retroviral production, together with 4 μg/mL of polybrene, were performed on MM13 and MM27 PDX cells. The 293T cell line was transfected with LLPG constructs for lentiviral production using calcium phosphate with pMD2.G-VSVG and pCD-NLBH packaging plasmids (generous gifts from Prof. Colin Goding). One cycle of infection (16 h) with lentiviral vectors, with 4 μg/mL of polybrene, was performed on MM13 and MM27 PDX cells and on WM115 and WM266.4 cell lines. Cells were puromycin (2 μg/mL) selected for 3 days and the surviving cells were used for subsequent experiments.

#### 4.1.3. siRNA Transfection Protocol

MM27 PDX cells were transfected following the DharmaFECT transfection protocol (Dharmacon Horizon) with ON-TARGET plus Human DOCK4 (9732) siRNA-SMARTpool (L-017968-01-0005, Dharmacon), ON-TARGET plus Human Rac1 (5879) siRNA-SMARTpool (L-003560-00-0005, Dharmacon) and ON-TARGET plus Non-targeting Pool (D-001810-10-05, Dharmacon). Briefly, the cells were transfected with siRNA-SMARTpool at the final concentration of 25 nM and DharmaFECT for 48 h.

#### 4.1.4. Cell Cultures

WM115 and WM266.4 melanoma cell lines were maintained in RPMI (Euroclone, Cat#ECM2001) supplemented with 10% FBS, 200 mmol/L glutamine, 100 U/mL penicillin and 100 μg/mL streptomycin. MM13 and MM27 PDX cells were maintained in Iscove’s modified Dulbecco’s medium (IMDM, Sigma Aldrich-Merck, Cat#I3390) supplemented with 10% North American origin FBS, 200 mmol/L glutamine, 100 U/mL penicillin and 100 g/mL streptomycin.

### 4.2. PDX Generation and In Vivo Studies

PDX were generated as previously described [21]. All the in vivo studies were performed after approval from our fully authorized animal facility and the notification of the experiments to the Ministry of Health (as required by the Italian Law; IACUCs N° 29/2013, N° 326/16 and N° 758/2015-PR), and in accordance with EU directive 2010/63.

Then, 7 × 10^5^ MM13 and MM27 PDX cells transduced with PP and PP-ShcD were transplanted by intradermal injection into 34 NOD.Cg-Prkdcscid Il2rgtm1Wjl/SzJ (NSG) mice. Mice were monitored for tumor development and excised when they reached ∼0.5 cm^3^ in volume (calculated using the modified ellipsoid formula ½ (length × width^2^)). Mice were then monitored weekly for metastasis formation by lymph node detection. After 1–2 months from resection, all mice were sacrificed and enlarged lymph nodes collected, eventually together with lung, liver, kidney and spleen. The lymph node volume was measured, with the same formula applied to primary tumors, as evidence of metastatic burden. All the organs were analyzed for the presence of macroscopic metastatic lesions.

### 4.3. Cell Adhesion and Spreading Assays

For the adhesion assay, ShLuc and ShShcD MM27 cells (2.5 × 10^5^ cells/well, 6-well-plate format, in duplicate) were plated onto different extracellular matrices: Fibronectin 5 μg/mL (Roche, Cat#11080938001), Collagen Type I Rat Tail 20 μg/mL (Corning, Cat#354249), Matrigel 20 μg/mL (Corning, Cat#356231). After 120 (Fibronectin), 100 (Collagen) and 80 (Matrigel) minutes, cells were fixed and stained with 0.5% Crystal Violet (Sigma Aldrich-Merck, Cat# V5265). Five images per well were acquired and images were analyzed (cell number) using ImageJ Software.

For the spreading assays, 2.5 × 10^4^ shLuc and shShcD cells were plated in triplicates on a 5 μg /mL fibronectin-coated (1 h at 37 °C) 24-well plate. After 75 min, the cells were fixed and stained with 0.5% Crystal Violet. Three images per well were acquired at EVOS XL CORE microscope, Life Technologies (20× and 40× magnification) and cell morphology and numbers were analyzed and counted using ImageJ Software.

### 4.4. Cell Morphology Assessment

In addition, 2.5 × 10^5^ MM27 PP and PP-ShcD cells/well (6-well-plate format) were plated on a thick collagen layer (2.3 mg/mL, Collagen Type I Rat Tail 20 μg/mL). Three images per well were acquired and the cells were counted as rounded or elongated on the basis of their shape. Experiments were carried out in triplicate.

The 2.5 × 10^5^ WM115 shLuc and shShcD#1 or #2 cells/well (6-well-plate format) were plated as previously described and treated for 24 h with 50 μM NSC23766 Rac1 inhibitor (Sigma Aldrich-Merck, Cat#SML0952). Five images per well were acquired and the cells were counted as described above.

### 4.5. Cell Migration Assay

The migration assay was performed using 8.0 μm pore size, fibronectin-coated inserts in 24-well plates (Corning-Falcon cat#353097). Triplicates of 3 × 10^4^ MM27 shLuc and ShcD-silenced cells were plated in the upper chamber in serum-free medium. Vehicle (DMSO) or Dabrafenib 2.5 nM (GSK2118436A, Active Biochem, Cat#A-1220), Trametinib 0.5 nM (GSK-1120212, Active Biochem, Cat#A-1258), and Everolimus 2.5 μM (RAD001, MedChemExpress, Cat#HY-10218) were added alone or in combination (Dabrafenib/Trametinib and Dabrafenib/Everolimus, same doses as single testing) to the medium. The complete medium was added to the lower chamber. After 48 h, cells that migrated to the lower surface of the inserts were stained with 0.5% Crystal Violet. Four images of each insert were acquired and analyzed with the ImageJ Software.

### 4.6. Spheroid Invasion Assay and Time-Lapse Analysis

For the spheroid formation, 2 × 10^3^ PP and PP-ShcD MM27 cells were grown as hanging drops in IMDM complete medium (Sigma Aldrich-Merck, Cat#I3390) with 0.4% methylcellulose for 48 h at 37 °C. Single spheroids were harvested, resuspended in 1.5 mg/mL Collagen Type I Rat Tail and plated on a 96 well-plate pre-coated with a thin collagen layer. After 1 h at 37 °C, the solidified collagen gel was covered with complete IMDM medium. The ability of cells to invade the area was monitored and imaged for 24 h on a TIRF Leica DMI6000B camera, temperature controller (37 °C), 5% CO_2_ incubation chamber (Leica), Okolab incubator and LAS AF Software. Images of 10 spheroids per group were collected every 15 min for 24 h using a dry 4× objective lens, and phase-contrast optics with a Z-stack ± 45 µm and 15 µm step. The invasion area was analyzed with the ImageJ Software and normalized against the area at time 0. The speed of the cells (μ/min) at the invasive front was analyzed by the manual tracking plugin of the ImageJ Software and calculated as a mean of 10 cells’ tracking. The rounded vs. elongated cell shape at the front of the invasion was carried out by the manual counting of 3 different fields of 5 spheroids per group on the ImageJ Software.

### 4.7. Immunofluorescence Staining

Furthermore, 2.5 × 10^4^ MM27 and WM115 cells infected with shLuc and shShcD were plated on 5 μg/mL fibronectin pre-coated (O/N at 4 °C) slides and allowed to attach for 75 min in complete IMDM. Cells were fixed with 4% paraformaldehyde for 10  min, permeabilized with 0.1% Triton-X, blocked for 1 h with 5% bovine serum albumin and immune-stained with Vinculin (Sigma Aldrich-Merck, Cat#V9131), p-Vinculin (Tyr-1065) (Invitrogen, Cat#44-10786), Paxillin (Invitrogen, Cat#03-6100), p-Paxillin (Tyr118) (Invitrogen, Cat# 44-7226), FAK (c-20, Santa Cruz, Cat#sc-558), p-FAK (Tyr397) (Upstate, Cat#07-012), DOCK4 (Novus Biologicals, Cat#NBP1-266-48) and CD146 (Novocastra, Cat#NCL-L-U). Secondary detection was done using secondary fluorescently labeled antibodies (Life Technologies Italia) for 45 min at room temperature. Slides were counterstained with 4′,6-diamidino-2-phenylindole (DAPI, Life Technologies Italia) for nuclei labelling and mounted on glass with Mowiol (Merk Life Science, Cat#81838). Images were collected at 63× magnification by motorized Leica DM6B fluorescence microscope, equipped with a Zyla camera, LASX Software, and by a Leica TCS SP5II confocal microscope, equipped with a Leica camera, LASX Software.

### 4.8. Immunoblot and Immunoprecipitation

For the immunoblot analysis, cells were rinsed with cold PBS and lysed in RIPA lysis buffer (150 mM sodium chloride, 1% NP-40, 0.5% sodium deoxycholate, 0.1% SDS, 50 mM Tris pH 8.0) plus protease and phosphatase inhibitor cocktail (Roche, Cat#11697498001 and ThermoScientific Cat#A32957), and sonicated twice for 20 s. Cell debris was removed by centrifugation at 13,000 rpm for 20 min at 4 °C. Protein concentration was determined using the Bradford Assay (Bio-Rad, Cat#500-0006). Protein lysates were separated on SDS-PAGE gel and transferred onto nitrocellulose membrane. Membranes were blocked with 5% nonfat milk or BSA in Tris-buffered saline with 0.1% Tween 20 (TBST) for 1 h. Membranes were immunoblotted with the following appropriately diluted primary antibodies overnight at 4 °C or 1 h at room temperature: ShcD anti-CH2 Clone 47-10-21 (home-made, see below), AXL H3 (Santa-Cruz, Cat#sc-166269), Sox10 (D5V9L) (Cell Signaling, Cat#89356), EGFR (E114) (Abcam, Cat#ab32562), EphA2 (D4A2) (Cell Signaling, Cat#6997), Brn2/POU3F2 (D2C1L) (Cell Signaling, Cat#12137), MITF antibody (C5) (Abcam, Cat#ab12039), PDGF-Rβ (R&D Systems, Cat#AF385), p-(Tyr-256) N-WASP (ECM Biosciences, Cat#WP2601), N-WASP (Novus Biologicals, #CatSC66-05), P-(Ser3)-Cofilin (Cell Signaling, Cat#3311), Cofilin (D3F9) (Cell Signaling, Cat#5175), ß-Actin (Novus Biologicals, Cat#NB600-503), Vinculin (Sigma Aldrich-Merck, Cat#V9131), GAPDH (Cell Signaling, Cat#2118). Membranes were washed three times with TBST and incubated with the appropriate horseradish peroxidase-conjugated secondary antibody for 1 h at room temperature. Finally, the expression of protein was detected by enhanced-chemiluminescence solutions (ECL-BioRad, Cat#170-5061) and captured by Amersham Hyperfilm (Cat#GEH28906837) or ChemiDoc XRS.

For immunoprecipitation, cells were harvested by scraping in PLC lysis buffer (50 mM HEPES pH 7.5, 150 mM NaCl, 10% glycerol, 0.5% Triton X-100, 1.5 mM MgCl2, 1 mM EGTA) plus protease and phosphatase inhibitor cocktail. Cells were lysed for 30 min on ice, centrifuged at 13,000 rpm for 30 min and the protein concentration was detected as above. Cell lysates (3 mg) were incubated with anti-DOCK4 Antibody (Novus Biologicals, Cat#NBP1-26648) for 3 h at 4 °C followed by 1 h incubation with protein A Sepharose CL-4B beads (GE-Healthcare, Cat#GE17-0780-01). As the negative control, cell lysates were immunoprecipitated with anti-IgG antibody. Immunocomplexes were washed 5 times with lysis buffer and boiled with sample buffer. Immunoblot analysis was performed with the indicated antibodies.

### 4.9. ShcD Antibody Generation (Anti-CH2 Clone 47-10-21)

The ShcD antibody was generated in collaboration with the Biochemistry Unit at the IFOM-European Institute of Oncology (IEO) campus. Antibodies against the CH2 domain of murine ShcD was raised using the peptide sequence of 16 amino acids: QPYRKYDNTGLLPPKK. This synthesized peptide sequence corresponds to the carboxy-terminal part of the CH2 region, which is specific for ShcD and not present in the other three members of the family. The immunized sera were tested on RaLP overexpressing cell lines by Western blotting and the reactive sera were immunopurified.

### 4.10. RAC1 Activation Assay

Rac1 activity was measured with Rac1 Activation Assay Biochem kit (Cytoskeleton, Cat#BK035) according to the manufacturer’s protocol. Cells were transduced with shLuc, shShcD#1 and shShcD#2 and plated on 5 μg/mL fibronectin-coated plates (as described above) for 15 min in complete RPMI medium. Cells were first washed with phosphate-buffered saline (PBS) and then lysed with a proper volume of lysis buffer provided by the manufacturer and supplemented with protease inhibitors. The lysates were precleared and after measuring protein concentration with the provided Precision Red Advanced Protein Assay, and 50 μg of lysates were saved for Western quantitation of total Rac1. About 800 μg of lysate was incubated with 15 μL of GST–PAK–PBD (Rac1 effector protein, p21 activated kinase 1) for specific pull-down of activated Rac1. After rocking at 4 °C for 1 h, beads were washed once and boiled at 95 °C for 2 min. Protein lysates were subsequently resolved in 14% SDS-PAGE gel and transferred to polyvinylidene fluoride membrane (PVDF).

The total and activated Rac1 was detected by Western blotting using an anti-Rac1 monoclonal antibody provided in the kit, followed by incubation with goat anti-mouse secondary antibody at room temperature for 1 h. Immunoblotting and detection were carried out as previously indicated.

### 4.11. Tissue Microarrays (TMA)

Tissue Microarrays were performed in collaboration with the Division of Pathology and the Molecular Pathology Unit at the European Institute of Oncology (IEO). Human specimens derived from formalin fixed and paraffin embedded melanocytic lesions were arrayed as previously described [43]. The choice of the samples was based on tumor availability but also on reliable and sufficient clinical information. Briefly, for each sample, two 0.6 mm cylinders from previously identified on hematoxylin-eosin stained sections were removed from the donor blocks and deposited on the recipient block using a custom-built precision instrument (Tissue Arrayer, Beecher Instruments, Sun Prairie, WI, USA). Two-micrometer sections of the resulting recipient block were cut, mounted on glass slides, and processed for in situ hybridation (ISH), as previously reported [44]. TMAs contained 183 primary melanomas, subdivided according to TNM staging and 94 metastatic melanomas. Patients were divided according to the diagnosis and ShcD status evaluated in primary and metastatic tissues. Patients carrying primary melanomas had a median age of 56 years at diagnosis, 97 were female (54%) and 83 were male (46%). The majority of primary melanomas were SSM (skin-sparing mastectomy) (68%, n = 125 patients, pts), thin (51%, n = 91 pts Breslow < 1 mm), non-ulcerated (72%, n = 131 pts) and with a mitotic rate ≥1 mm^2^ (62%, n = 114). Lymph node status was positive in 35 patients (19%) and not available in 12 patients (7%). In the metastatic group, 68% of the patients were male. Fifteen patients (15%) had an occult primary melanoma, while for the remaining 79 patients with a known antecedent primary melanoma, some but not all primary tumor parameters were available. The majority of the melanomas were thick (75% with Breslow > 1 mm), with a high mitotic rate (51% with mitoses >1 mm^2^) and a positive lymph node status (49% positive, 28% negative and 23% unknown). The sites of primary tumor were often trunk (43% and 41% for primary and metastatic tissue, respectively) and extremities (45% and 32% for primary and metastatic tissue, respectively).

### 4.12. In Situ Hybridization (ISH)

ShcD expression levels were assessed by ISH as previously reported [44]. TMA sections were deparaffinized, digested with Proteinase K (20 μg/mL), postfixed, acetylated and dried. After overnight hybridization at 50 °C, the sections were washed in 50% formamide, 2 × SSC, 20 mM 2-mercaptoethanol at 60°, coated with Kodak NTB-2 photographic emulsion and exposed for three weeks. The slides were lightly H&E counterstained and analyzed under microscope with a dark-field condenser for the silver grains.

All TMA were first analyzed for the expression of the housekeeping gene β-actin, against which the specific signals were normalized. Gene expression levels were evaluated by three independent operators, counting the number of grains per cell and were expressed in a semi-quantitative scale (ISH score): 0 (no staining), 1 (1–25 grains; weak staining), 2 (26–50 grains; moderate staining), and 3 (>50 grains, strong staining). ISH scores 2 and 3 were considered to represent an unequivocal positive signal.

### 4.13. Drug Treatment

In vitro drug sensitivity was assessed by CyQuant (Invitrogen, Cat#C35012) in MM27 shLuc and shShcD cells. Briefly, the cells were plated in triplicate in 96 wells (1500 cells/well) and treated for 72 h by a single exposure to either the vehicle or increasing concentrations of Dabrafenib (BRAF inhibitor, in the range of 5–100 nM), Trametinib (MEK inhibitor, in the range of 0.1–5 nM), Everolimus (mTOR inhibitor, in the range of 1–25 μM). The inhibition of viability is indicated as a percent over control (GraphPad Prism software).

### 4.14. Statistical Methods

Clinical and pathologic features were tested for association with ShcD status using simple cross tabulations, chi-square test, Mantel–Haenszel and Fisher’s exact test; median values and range inter-quartiles were presented for continuous variables and compared with non-parametric Wilcoxon tests.

Time to death and time to recurrence were defined as the time from the first diagnosis of melanoma until the event of interest. For the analyses among metastatic patients with ShcD evaluated in the metastatic tissue, time to death was defined as the time from the diagnosis of first distant metastasis until death. All patients, alive or free of disease at last follow-up date, were considered right censored. Multivariable binary logistic regression was used to assess the statistically significant features associated with the ShcD status. Disease-free survival was estimated by the Kaplan–Meier method to evaluate the time to events. The log-rank test was used to compare the survival time between groups. Cox proportional hazards models were used to assess if the ShcD was associated with survival, after adjustments for confounders (age and gender), for disease-free survival and also chemotherapy for overall survival. Prognostic factors, such as Breslow thickness, ulceration and lymph node involvement, were included, if significant, in multivariate models for metastatic tissue. In primary tissue, they were not included because they were considered intermediate steps in the casual path between ShcD and recurrence or death. In fact, we hypothesized that ShcD had either a direct influence on cell invasion or it indirectly regulated target molecules that lead to reduced invasive potential.

All statistical tests were two-sided, and *p* < 0.05 was considered statistically significant. The statistical analyses were performed with the Statistical Analysis System Version 9.2 (SAS Institute, Cary, NC 27513, USA).

All in vitro and in vivo data were represented as the mean ± SD of biological triplicates (if not diversely indicated in the text). Comparisons between two or more groups were assessed by using two-tailed Student’s *t* test. *p* < 0.05 and lower were considered significant.

## 5. Conclusions

Comprehension of how melanoma infiltrates the surrounding tissues and disseminate is key to designing new and more effective therapeutic strategies. The metastatic cascade is a multi-step process still not completely dissected. Our results indicate that ShcD, a signaling adaptor protein, drives changes in melanoma plasticity, enabling cells to switch between different types of motility and therefore invade and metastasize. The uncovering of this novel signal transduction pathway may contribute to the understanding of how metastases develop and evolve, and how the different phenotypes leading to metastasis can be either reversed or efficiently inhibited. Our results may open new therapeutic perspectives.

## Figures and Tables

**Figure 1 cancers-12-03366-f001:**
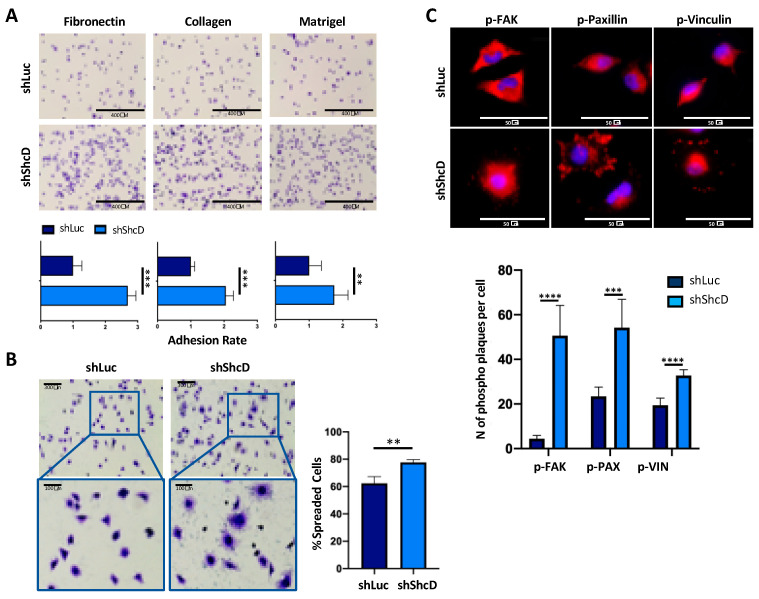
ShcD depletion modifies the adhesive phenotype of melanoma cells. (**A**) ShLuc and ShShcD (pool of ShShcD#1 and #2) MM27 cells’ adhesion to different extracellular matrices (fibronectin, collagen, matrigel). Cells were detected by Crystal Violet staining. Adhesion rate is calculated as the ratio of relative numbers of adherent cells in shShcD vs. shLuc. Five images per well were acquired in duplicate experiments. Student *t-test* (***, *p* < 0.001; **, *p* < 0.01, ****, *p* < 0.0001) was applied to assess the significance. Representative images are shown (20×). (**B**) ShLuc and ShShcD MM27 cells spreading evaluation on fibronectin. Cells were stained with Crystal Violet. Images were quantified with ImageJ software. Data are shown as the mean ±SD of 3 fields of 3 different cover slips. Student *t-test* (**, *p* < 0.01). (**C**) ShLuc and ShShcD MM27 cells’ focal adhesion analysis by immunofluorescence. Cells were treated as in (**B**) and the protein expression of p-vinculin, p-paxillin and p-FAK (red) was detected. Nuclei were counterstained with DAPI (blue). Representative images are shown (63× magnification).

**Figure 2 cancers-12-03366-f002:**
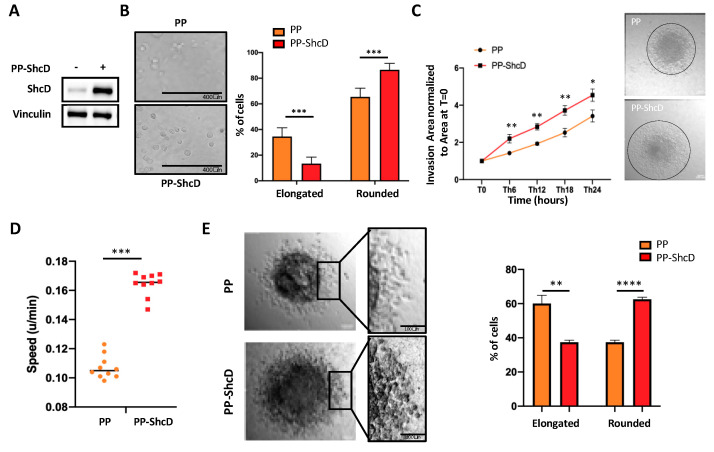
ShcD overexpression increases the invasion and amoeboid movement of melanoma cells. (**A**) ShcD overexpression was estimated by immunoblot analysis in MM27 cells transduced to overexpress ShcD (PincoPuro (PP)-ShcD) or a neutral control (PP, empty vector). Vinculin was used as a loading control. (**B**) PP/PP-ShcD MM27 cell morphology assessment. Cells were plated on a thick collagen layer and the percentage of elongated/rounded cells was calculated after 24 h. Three images per well were acquired in triplicate experiments. Student *t-test* (***, *p* < 0.001). (**C**) Spheroid collagen invasion assay in PP and PP-ShcD MM27 cells. The area of invasion was monitored by time-lapse microscopy for 24 h. Data are shown as the mean ±SD of 10 different spheroids per group. Student *t-test* (*, *p* < 0.05; **, *p* < 0.01). Representative images are shown. (**D**) Spheroid invasion speed in PP and PP-ShcD MM27 cells at the front of invasion. Cell speed was analyzed with the manual tracking plugin of the ImageJ software. Data are shown as the mean ±SD of 2 different cells of 5 spheroids per group. Student *t-test* (***, *p* < 0.0001). (**E**) Assessment of cell shape at the front of invasion in PP and PP-ShcD MM27 cells. Cell shape was analyzed and quantified by scanning 2 high-power fields of 5 spheroids per sample using ImageJ software. Student *t-test* (**, *p* < 0.01; ****, *p* < 0.0001). Representative images are shown.

**Figure 3 cancers-12-03366-f003:**
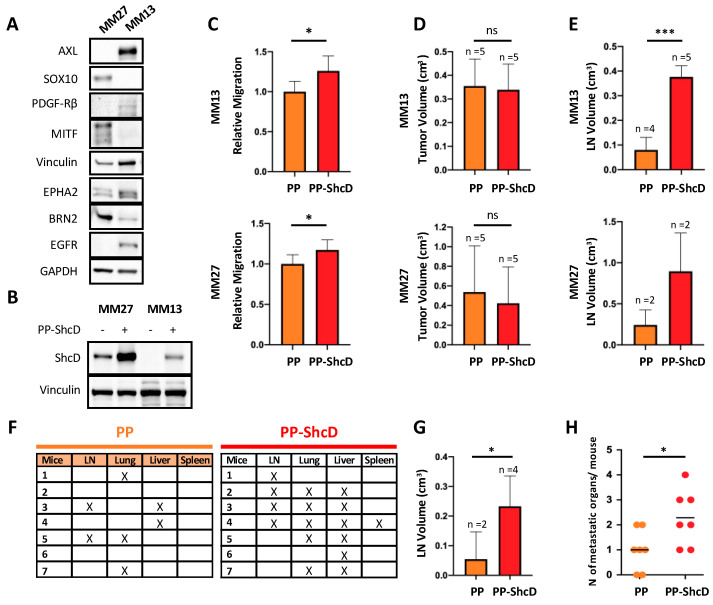
ShcD overexpression promotes metastasis formation at lymph nodes and distant sites. (**A**) Phenotype characterization of MM27 and MM13 patient-derived xenografts (PDXs) cells by Western blotting. Immunoblots show the protein expression of invasive markers (AXL, PDGFR-β, EGFR, EPHA2) and proliferative markers (SOX10, MITF, BRN2). Vinculin and GAPDH were used as loading controls. (**B**) The assessment of ShcD overexpression in MM27 and MM13 PDX cells by Western blot analysis, before mice transplantation. MM27 and MM13 PDX cells were transduced to overexpress ShcD (PP-ShcD) or a neutral control (PP, empty vector). Vinculin was used as a loading control. (**C**) Transwell migration assay in PP/PP-ShcD MM13 and MM27 cells, at the 48 h time point. Relative migration (mean ± SD) is expressed as a ratio of PP-ShcD vs. PP cell migration values (calculated with ImageJ). Student *t*-test (*, *p* < 0.05). (**D**–**H**) In vivo metastatic potential of PP and PP-ShcD MM13 and MM27 cells. Transduced cells were intradermally injected in NSG mice (D–E n = 5 per group; F–H MM13 n = 8 per group) and the tumors were surgically resected. The early time point evaluation of metastatic dissemination (MM13, MM27): (**D**) tumor volumes at resection. Student *t-test* (not significant, n.s.). (**E**) Lymph nodes (LNs) volumes at mouse sacrifice. MM13 PP, n = 4; PP-ShcD, n = 5; MM13 PP, n = 2; PP-ShcD, n = 2. Student *t-test* (***, *p* < 0.0001). Late time point evaluation of metastatic dissemination (MM13): **F**) metastatic organs evaluation of MM13 PP and PP-ShcD injected mice. “X” indicates presence of metastatic foci in the reported organs (n = 7). (**G**) Lymph nodes’ (LNs) volumes at mouse sacrifice. MM13 PP, n = 2; PP-ShcD, n = 4. Student *t-test* (*, *p* < 0.05). (**H**) Number of metastatic organs (LNs and distant metastasis) per mouse in PP and PP-ShcD MM13 injected mice. PP, n = 7; PP-ShcD, n = 16. Student *t-test* (*, *p* < 0.05).

**Figure 4 cancers-12-03366-f004:**
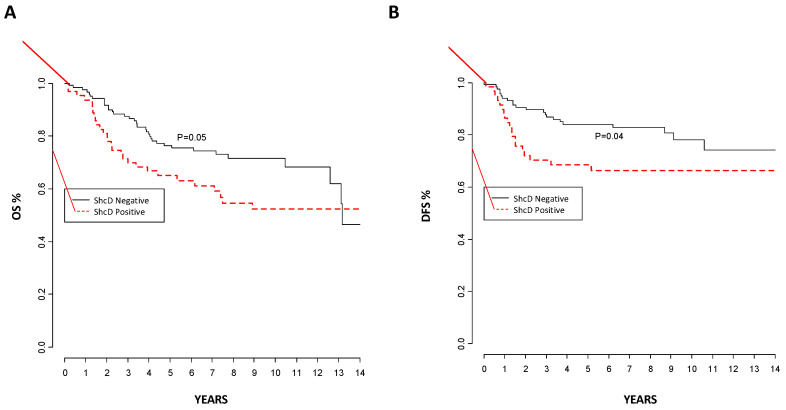
ShcD expression correlates with melanoma progression and poor prognosis in melanoma patients. (**A**) ShcD-positive patients show a significant worse overall survival than the ShcD-negative patients (log-rank *p* = 0.05). (**B**) Disease-free survival at two years is significantly worse (66%) in ShcD-positive patients than in ShcD-negative patients (78%; log-rank test *p* = 0.04, stage IV patients not included in the analysis).

**Figure 5 cancers-12-03366-f005:**
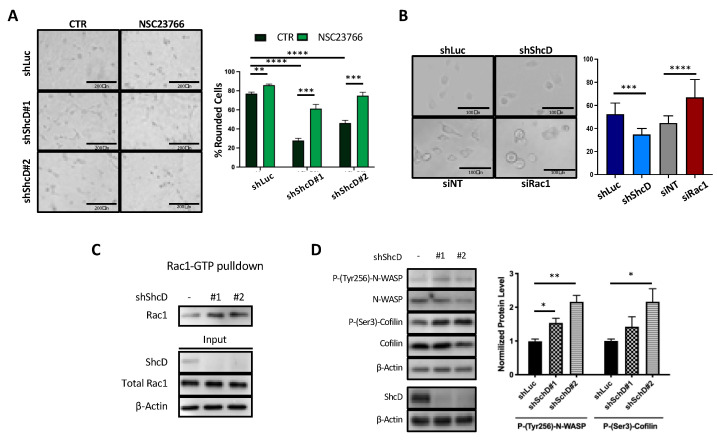
ShcD-silencing cells show a deregulated Rac1 molecular pathway. (**A**) shLuc and shShcD WM115 cells morphology assessment upon Rac1 inhibitor treatment (NSC23766). Cells were transduced with shLuc, shShcD#1 and shShcD#2, plated on a thick collagen layer and treated with either vehicle (CTR, control, DMSO) or 50 μM NSC23766. After 24 h treatment, the percentage of elongated/rounded cells was calculated. Three images per well were acquired in triplicate experiments. Student *t-test* (***, *p* <  0.001; ****, *p*  <  0.0001). (**B**) MM27 cells morphology assessment upon Rac1 silencing. Cells were infected with shLuc and shShcD (pool of shShcD#1 and #2) or transfected with the non-targeting and Rac1 SmartPool siRNA and after 48 h plated on a thick collagen layer. After 24 h, the percentage of elongated/rounded cells was calculated. Three images per well were acquired in triplicate experiments. Student *t-test* (****, *p* <  0.0001). (**C**) Rac1 activation assay in shLuc and shShcD WM115 cells. Rac1 activity was assayed by GTP-bound Rac1 pulled-down and analyzed by Western blotting. WM115 cells were transduced with shLuc, shShcD#1 and shShcD#2 and plated on fibronectin. Rac1 indicates active Rac1 in the pull-down assay, and Total Rac1 represents the total amount of Rac1 in the cell lysates (input). Actin was used as a loading control. (**D**) Rac1 pathway regulation upon ShcD silencing in WM115 cells. WM115 cells were transduced with shLuc, shShcD#1 and shShcD#2 and plated on fibronectin. The Rac1 molecular pathway was investigated at the protein level by Western blotting in ShcD-depleted WM115 cells. Both the phosphorylation status of N-WASP (P-Tyr256) and Cofilin (P-Ser3) and their total expression levels were detected and ShcD silencing was also checked. Actin was used as a loading control. The quantification of two experiments was performed with ImageJ normalizing for the total protein level and house-keeping gene. Student *t-test* (*, *p* < 0.05; **, *p* < 0.01).

**Figure 6 cancers-12-03366-f006:**
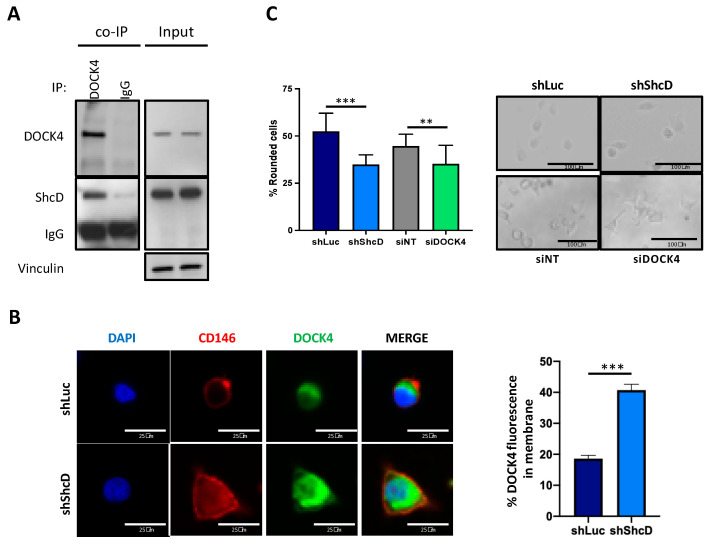
Rac1 molecular pathway is altered in ShcD expressing melanoma cells via DOCK4. (**A**) DOCK4 immunoprecipitation in ShcD overexpressing MM27 cells. Cell lysates of PP-ShcD MM27 cells were immunoprecipitated (IP) with anti-DOCK4 antibody or anti-IgG antibody and subsequently immunoblotted with anti-ShcD antibody. Expression of vinculin was used as a loading control in total cell lysates (input). (**B**) DOCK4 localization in shLuc and shShcD MM27 cells. Cells were plated on fibronectin and co-stained for DOCK4 and CD146 plasma membrane marker by immunofluorescence (red). Nuclei were counterstained with DAPI (blue). Representative images are shown (63× magnification). Quantification was done using ImageJ and the DOCK4 plasma membrane signal was normalized to the total fluorescence. Student *t-test* (***, *p* < 0.001). (**C**) The analysis of cell morphology in DOCK4 siRNA and ShShcD-silenced MM27 cells. DOCK4-silenced cells and their controls together with ShLuc and ShShcD cells were plated on a thick layer of collagen and rounded vs. the elongated morphology assessed after 24 h. The percentage of elongated/rounded cells was calculated. Three images per well were acquired in triplicate experiments. Student *t-test* (**, *p* < 0.01).

**Figure 7 cancers-12-03366-f007:**
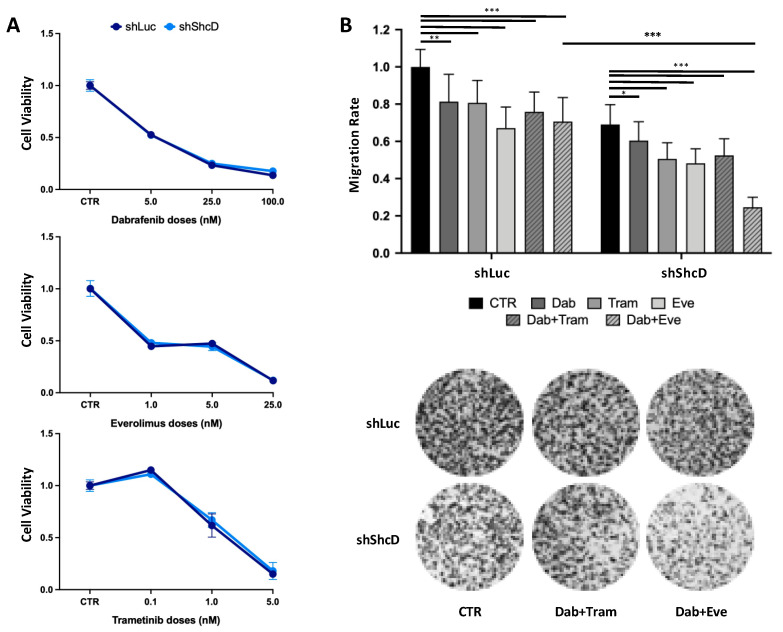
ShcD silencing sensitizes melanoma cells to targeted therapy combination. (**A**) ShLuc and ShShcD MM27 cell viability upon treatment with Dabrafenib (BRAF inhibitor), Trametinib (MEK inhibitor) and Everolimus (mTOR inhibitor). Cells were treated for 72 h and cell viability was assessed by CyQuant. The inhibition of viability is indicated as a percent over control (GraphPad Prism software). Student *t*-test (not statistically significant). (**B**) Transwell migration assay upon specific targeted therapy in ShLuc and ShShcD MM27 cells. Cells were plated in the upper chamber in a serum-free medium with Dabrafenib (2.5 nM), Trametinib (0.5 nM), Everolimus (2.5 μM), Dabrafenib/Trametinib and Dabrafenib/Everolimus combinations (same doses used for the single drug testing) or vehicle (CTR, DMSO). Complete medium was added to the lower chamber. After 48 h, the cells were stained with Crystal Violet and the migration rate was quantified using ImageJ. Histograms represent the relative cell migration (mean ± SD) of a triplicate experiment, normalized to ShLuc as the control. Differences among the groups were calculated by applying the Student *t-test*, using migration rate values normalized to the CTR of the corresponding group (*, *p* < 0.05; ***, *p* < 0.001; **, *p* < 0.01).

**Table 1 cancers-12-03366-t001:** Patient demographics and clinical characteristics of the primary cutaneous melanoma according to the ShcD status (AJCC VII).

		ShcD in Primary Tissue				ShcD in Metastatic Tissue		
	No. of Patients, n (%)	Negative	Positive	*p*-Values	No. of Patients, n (%)	Negative	Positive	*p*-Values
No. of patients, n (%)	183 (100%)	120 (66%)	63 (34%)		94 (100%)	40 (43%)	54 (57%)	<001
**At diagnosis**					**Antecedent CM**			
*Sex*				0.46				
Female	97 (54%)	67 (56%)	30 (50%)		29 (31%)	13 (32%)	16 (30%)	0.76
Men	83 (46%)	53 (44%)	30 (50%)		65 (69%)	27 (68%)	38 (70%)	
*Age at diagnosis*				0.21				
Median	56	54	59		48	50	45	0.02
Interquartile range	(44, 67)	(43, 65)	(46, 69)		(32, 59)	(41, 62)	(24, 55)	
*Breslow thickness mm*				<0001				
Median	1.0	0.5	2.4		2.8	1.9	3.1	0.14
Interquartile range	(0.5, 3.1)	(0.3, 1.6)	(1.2, 4.7)		(1.5, 5.0)	(0.9,4.8)	(1.8, 5.4)	
*Breslow thickness mm*				<0001				0.12
0.01–1.0	91 (51)	80 (67)	11 (18)	(<0001 †)	9 (25)	7 (67)	2 (4)	(0.02)
1.01–2.0	26 (14)	14 (12)	12 (20)		20 (21)	8 (17)	12 (22)	
2.01–4.0	31 (17)	15 (13)	16 (27)		16 (17)	3 (20)	13 (24)	
>4.0	31 (17)	10 (8)	21 (35)		25 (27)	10 (8)	15 (28)	
Unknown	1 (1)	1 (1)	0 (0)		24 (1)	12 (30)	12 (22)	
*Site of primary*				0.75				0.23
Extremity	83 (45%)	51 (43%)	30 (50%)	(0.28 ‡)	30 (32%)	12 (30%)	18 (33%)	(0.73 ‡)
Trunk	78 (43%)	54 (45%)	23 (38%)		38 (41%)	16 (40%)	22 (41%)	
Head and neck	13 (7%)	9 (8%)	4 (7%)		6 (6%)	1 (3%)	5 (9%)	
Other	7 (4%)	4 (3%)	3 (5%)		5 (5%)	4 (10%)	1 (2%)	
Occults	2 (1%)	2 (1%)	0 (0%)		15 (16%)	7 (18%)	8 (15%)	
Ulceration				<0001				0.04
Yes	46 (25%)	14 (12%)	32 (51%)	(<0001)	32 (34%)	9 (23%)	23 (43%)	(0.04)
No	131 (72%)	102 (85%)	29 (46%)		23 (24%)	13 (32%)	10 (18%)	
Unknown	6 (3%)	4 (3%)	2 (3%)		39 (42%)	18 (45%)	21 (39%)	
Mitotic rate				<0001				0.27
Mitoses <1 mm^2^	64 (35%)	60 (50%)	4 (6%)	(<0001)	32 (34%)	16 (40%)	16 (30%)	(0.06)
Mitoses ≥1 mm^2^	114 (62%)	55 (46%)	59 (94%)		48 (51%)	18 (45%)	30 (55%)	
Unknown	5 (3%)	5 (4%)	0 (0%)		14 (15%)	6 (15%)	8 (15%)	
Histo-pathologic subtype				0.001				0.83
NM	53 (29%)	24 (20%)	29 (46%)	(0.0004¥)	12 (13%)	4 (10%)	8 (15%)	(0.63¥)
SSM	125 (68%)	93 (77%)	32 (51%)		37 (39%)	16 (40%)	21 (39%)	
Other	4 (2%)	2 (2%)	2 (3%)		5 (5%)	2 (5%)	3 (5%)	
Unknown	1 (1%)	1 (1%)	0 (0%)		40 (43%)	18 (45%)	22 (41%)	
*AJCC stage*				0.0004				0.33
I	105 (58%)	83 (69%)	22 (35%)	(0.01)	7 (7%)	4 (10%)	3 (5%)	(0.02§)
II	33 (18%)	16 (13%)	17 (27%)		19 (20%)	10 (25%)	9 (17%)	
III	31 (17%)	16 (13%)	15 (24%)		46 (49%)	18 (45%)	28 (52%)	
IV	6 (3%)	2 (2%)	4 (6%)		3 (3%)	0 (0%)	3 (5%)	
Unknown	8 (4%)	3 (3%)	5 (8%)		19 (20%)	8 (20%)	11 (21%)	
Lymph-nodes status				0.02				0.33
Positive	35 (19%)	17 (14%)	18 (28%)	(0.03)	46 (49%)	17 (43%)	18 (54%)	(0.02)
Negative	136 (74%)	94 (78%)	42 (67%)		26 (28%)	15 (37%)	11 (20%)	
Unknown	12 (7%)	9 (8%)	3 (5%)		22 (23%)	8 (20%)	14 (26%)	
**ShcD evaluation**								
First metastasis								0.01
Regional metastasis					76 (81%)	28 (70%)	48 (89%)	(0.94)
Distant metastasis					17 (18%)	12 (30%)	5 (9%)	
Unknown					1 (1%)	0 (0%)	1 (2%)	
**Events during follow-up**								
First recurrence *				0.04				
Yes	41 (23%)	22 (19%)	19 (32%)	(0.08)				
No	136 (77%)	96 (81%)	40 (68%)					
Deaths				0.05				0.32
Yes	64 (35%)	36 (30%)	28 (44%)	(0.39)	54 (57%)	11 (69%)	43 (55%)	(0.40)
No	119 (65%)	84 (70%)	35 (56%)		40 (43%)	5 (31%)	35 (45%)	

*p*-values from chi-square, Fisher exact, Mantel–Haenszel or Wilcoxon tests. In parentheses, *p*-values from the multivariate logistic model. For an event during follow-up, *p* = value from Log rank tests and among parentheses from multivariate Cox regression model. § stage III or IV vs. I or II; ¥ SSM (superficial spreading melanoma) vs. NM (nodular melanoma); ‡ trunk vs. other; † Breslow >1 mm vs. ≤1 mm; ***** any type of first recurrence among non-metastatic patients (stage IV excluded) and adjusted for age, gender, and ulceration.

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
