# Peer review of "ShcD Binds DOCK4, Promotes Ameboid Motility and Metastasis Dissemination, Predicting Poor Prognosis in Melanoma"

_cancers, 2020, doi:10.3390/cancers12113366_

Round 1
Reviewer 1 Report
The Authors have fully addressed the required issues. I believe that the paperi s interesting and well-organized,
The Authors have fully addressed the required issues. I do not have other criticisms. I believe that the paper is original and well-organized, it might be of interest for the readers of Cancers journal.
Reviewer 2 Report
Highly improved revised form. Undoubtedly and with pleasure suggested to be accepted for publication.
This manuscript is a resubmission of an earlier submission. The following is a list of the peer review reports and author responses from that submission.
Round 1
Reviewer 1 Report
​I have read with interest the paper by Aladowicz and coll, that is focused on the investigation of the effects of ShcD proteins on melanoma capability to invade tissues and metastatize.
I believe that the topic is original and the paper is interesting; furthermore the experimental plan is well documented/conducted. I do not have substantial criticisms related to the experimental part of the manuscript.
I have only one some comments that are listed above:
- section 2.2, line 132: “…by culturing cells in three-dimensional conditions.” I am wondering if organotypic cultures are available for melanoma setting. In such case, I would suggest to add a short paragraph to discuss it (and why the Authors did not perform their experiments by the organotypic cultures instead of 3D matrix);
- section 2.4, line 253: more ulcerated: I believe that it shoud be corrected with “ulceration was more frequently represented”;
- Table 1: it should be specified which AJCC classification was mentioned to classify the patients group (the VII or the VIII classification?). Maybe the group of cases is composed by patients enrolled before 2018 (when the VII classification was used) and after 2018. In such case, it shoud be reported in the Table legend;
- section 2.4, line 290: “suggesting that ShcD in the early stages of melanoma might predict for metastasis dissemination”. In my opinion, this could not be stated (early stages?), since in the whole group of patients, about one third was made by patients with metastatic disease. Moreover, the authors write “ShcD-positive patients …carrying more frequently ulcerated melanomas…”(lines 287-289): in general terms, the presence of ulceration in melanoma is not associated to early stage of the disease.
- section 2.6: this is a very interesting section, since it represents the translational part of this research. Did the Authors evaluate the ShcD levels in all the group of patients reported in Table 1? This is not clear, but could be very important, and the consequent correlation between the protein level/expression in tissues and clinical history could add further powerful data, thus potentially validate the experimental results and allowing their translation to clinical research. So, the data from Table 1 and Figure 4 should be better discussed. Hence, the text in section 2.6 seems to be related only to in vitro experiments, while I strongly suggest to integrate it with the clinical part of this project.
Furthermore, do the Authors believe that the melanoma immunology treatment (i.e. nivolumab, pembrolizumab) could play a role in ShcD protein expression (any data in the literature?)
- Finally, I am sorry but I was not able to find the link to MovieS1.
Reviewer 2 Report
This is an experimentally sound and interesting manuscript that merits publication in "Cancers" - MDPI journal, providing the following suggestions to be successfully addressed by the authors.
1. Data of Fig. 1C have to be confirmed by Western blotting. Quantification is also required. Authors are additionally advised to examine the levels of phosphorylated FAK, phosphorylated Paxillin and phosphorylated Vinculin, instead of only their total protein contents. It may the elevated phosphorylation of focal-adhesion components that is associated with ShcD lack/downregulation.
2. What is the percentage of rounded and elongated cells in WM266.4?
3. Fig. 2E has to be replaced. It does not convincingly support the conclusions. It seems that the number of round/amoeboid cells is increased in control (PP) as compared to ShcD-overexpressed conditions, which argues against authors' statements. The overall size of rounded cells also seems different between control and ShcD+ cells. Authors should comment on that.
4. Fig. 5C is not convincing. The #1 and #2 shShcD do not show the same results, regarding the phospho-WASP and phospho-Cofilin levels increase (even if they are compared to their respective total protein contents). How many times was the experiment repeated? Authors are advised to re-draw the Figure with new and more concise experimental results.
5. Why are there two (2) β-Actin controls in Fig. 5C? Do they represent two (2) different experiments. Authors are advised to remove the one (1) control, accordingly.
6. Fig. 5C has to be quantified. The same applies for Fig. 5E. What is the percentage of DOCK4 that is recruited to the plasma membrane? A higher magnification of the respective image illustrating the membrane topology of DOCK4 is required.
7. DOCK4 RNAi-mediated silencing is suggested to be conducted and next compared with the ShcD RNAi-obtained cellular phenotypes. The same applies for Rac1, since a Rac1 RNAi-induced suppression may likely phenocopy the NSC23766 responses, thus strengthening the molecular model's value.
8. DOCK4 has to be removed from the title. Only two (2) experiments (Figs. 5D and 5E) cannot support its position in manuscript's title.
9. A Reference has to replace the phrase "... data from ... on Cancer, Lyon ...", line 54.
10. Scale bars must be added in all images (e.g. Fig. 1A, Fig. 1B, Fig. 1C, Fig. 2B, Fig. 2C, Fig. 2E, Fig. 5A and Fig. 5E).
11. Line 156: "Figure 2E" has to be replaced by "Figure 2D".
12. Lines 243 - 255: Since the melanoma patient features are described in Table 1 (must be shown in bold fonts), they can be removed (or, size-reduced) from the main text body of the manuscript.
13. Lines 298 - 301: Font size has to be reduced.
14. The whole text, including Figure Legends, has to be thoroughly revised, in terms of grammar, syntax, spelling and punctuation mistakes, by a Native speaker of the English language, who also happens to be an expert in the particular field of scientific interest.